

# Comparisons of spectral aerosol absorption in Seoul, South Korea

Jungbin Mok[1,2], Nickolay A. Krotkov[2], Omar Torres[2], Hiren Jethva[2,3], Zhanqing Li[1], Jhoon Kim[4], Ja-Ho Koo[4], Sujung Go[4], Hitoshi Irie[5], Gordon Labow[2], Thomas F. Eck[2,3], Brent N. Holben[2], Jay Herman[2], Robert P. Loughman[6], Elena Spinei[1,2], Seoung Soo Lee[1], Pradeep Khatri[7] and Monica Campanelli[8]

[1]Earth System Science Interdisciplinary Center (ESSIC), College Park, Maryland, USA
[2]NASA Goddard Space Flight Center, Greenbelt, Maryland, USA
[3]Universities Space Research Association, Columbia, Maryland, USA
[4]Institute of Earth, Astronomy, and Atmosphere, Brain Korea 21 Program, Department of Atmospheric Sciences, Yonsei University, Seoul, Republic of Korea
[5]Center for Environmental Remote Sensing, Chiba University, Chiba, Japan
[6]Department of Atmospheric and Planetary Sciences, Hampton University, Hampton, Virginia, USA
[7]Center for Atmospheric and Oceanic Studies, Graduate School of Science, Tohoku University, Sendai, Japan
[8]Consiglio Nazionale delle Ricerche (CNR), Institute of Atmospheric Sciences and Climate (ISAC), Rome, Italy

*Correspondence to*: Jungbin Mok (jungbin.mok@nasa.gov)

**Abstract.** Quantifying the aerosol absorption at ultraviolet (UV) wavelengths is important for monitoring air pollution using current (e.g., Aura/OMI) and future (e.g., TROPOMI, TEMPO, GEMS, and Sentinel-4) satellite measurements. Measurements of column atmospheric aerosol absorption (i.e., column effective imaginary refractive index ($k$), single scattering albedo (SSA), and aerosol absorption optical depth (AAOD)) are performed on the ground by the NASA AERONET in the visible (VIS) and near-infrared (NIR) wavelengths and in the UV-VIS-NIR by the SKYNET networks. Previous comparison studies have focused on visible and NIR wavelengths due to the lack of co-incident measurements of aerosol and gaseous absorption properties in the UV. This study compares the SKYNET-retrieved SSA in the UV with the SSA derived from a combination of AERONET, MFRSR, and Pandora (AMP) retrievals in Seoul, South Korea in spring and summer of 2016. The results show that the spectrally invariant surface albedo assumed in the SKYNET SSA retrievals leads to underestimated SSA compared to AMP values at near UV wavelengths. Re-processed SKYNET inversions using spectrally varying surface albedo, consistent with the AERONET retrieval improves agreement with AMP SSA. The combined AMP inversions allow for separating aerosol and gaseous ($NO_2$ and $O_3$) absorption and provides aerosol retrievals from the shortest UVB (305 nm) through visible to NIR wavelengths (870 nm).



## 1 Introduction

Aerosols affect the surface and outgoing radiation perturbing Earth's radiative balance. To quantify the radiative effects of aerosols, the aerosol optical depth (AOD) and single scattering albedo (SSA) are monitored using ground-based, orbital and sub-orbital platforms. The aerosol absorption changes atmospheric stability and cloud formation (aerosol semi-direct effects

(Hansen et al, 1997; Johnson et al., 2004)) and modifies the hydrological cycle (Koren et al., 2004; Li et al., 2016; Lee et al., 2017). The potential climate effects of absorbing aerosols have received considerable attention lately (Myhre et al., 2013). Moreover, aerosol effects on surface ultraviolet (UV) irradiance and photolysis rates have important implications for tropospheric photochemistry, human health, and agricultural productivity (Dickerson et al., 1997; Krotkov et al., 1998; He and Carmichael, 1999; Castro et al., 2001; Mok et al., 2016). The photochemical reactions involving UV radiation produce

secondary pollutants including tropospheric ozone, which has negative effects on air quality and agricultural crop yield (Tong et al., 2007; Fishman et al., 2010). Long-term exposure to secondary pollutants leads to exacerbation of asthma and permanent lung damage (Pope et al., 2003; Jerrett et al., 2009). Quantifying the aerosol absorption at UV wavelengths is important for monitoring air pollution using current (e.g., Aura/OMI) and future (e.g., TROPOMI, TEMPO, GEMS, and Sentinel-4) satellite measurements (Torres et al., 1998; 2007; 2013). Measurements of column atmospheric aerosol

absorption and its spectral dependence in the UV remain one of the most difficult tasks in atmospheric radiation measurements.

The enhanced column UV absorption (lower SSA at wavelengths shorter than 440 nm) is commonly attributed to organic aerosols (OA) that absorb predominantly in the UV, explaining much stronger wavelength dependence than a purely black

carbon (BC) absorption would suggest (Kirchstetter et al., 2004). Martins et al. (2009) show that the absorption efficiency of urban aerosol is considerably larger in the UV than in the VIS wavelengths and is probably linked to the absorption by OA. This enhanced UV absorption by OA results in a doubling of absorption efficiency compared to BC alone and reduces surface UV fluxes by up to 50%. Mok et al. (2016) first measured enhanced UV absorption with the strong spectral dependence attributed to light absorbing component of organic carbon (OC) known as "brown carbon" (BrC) for aged

Amazonian biomass burning smoke. Although urban aerosols have different chemical and physical composition, they also exhibit larger UV absorption with significant impact on the tropospheric photochemistry and biologically active surface UV irradiance (Krotkov et al., 1998; 2005b; Li et al., 2000; Ciren and Li, 2003; Bergstrom et al. 2007; 2010; Arola et al., 2009; Mok et al., 2016).

Recently, the need for measurements of column atmospheric aerosol absorption in the UV wavelengths are highlighted in the global aerosol and chemistry transport model (CTM) simulations. Current CTMs treat all OC from biomass burning as purely scattering and thus underestimate heating effect of the total carbon (BC+OC) – primary absorbing component of carbonaceous aerosols (Cooke et al., 1999; Chung and Seinfeld, 2002; Bond et al., 2013; Myhre et al., 2013). However,



recent laboratory studies (Kirchstetter et al., 2004; Yang et al., 2009; Chakrabarty et al., 2010; Chen and Bond, 2010; Lack et al., 2012; Saleh et al., 2013; 2014; Zhong and Jang, 2014) suggest that BrC is capable of enhancing total absorption efficiency of OC, potentially altering the direct radiative forcing (DRF) from negative to positive (Bond, 2001; Kirchstetter et al., 2004; Feng et al., 2013; Saleh et al., 2014). Recently, Hammer et al. (2016) show that carbonaceous aerosol absorption

over most of biomass burning regions is underestimated if OC is regarded as purely scattering in a global 3-D CTM GEOS-Chem, while a better agreement is obtained with satellite observations from the Ozone Monitoring Instrument (OMI) on board NASA's Aura satellite after implementing the BrC absorption parameterization.

The aerosol column absorption in the VIS and NIR wavelengths is measured routinely in many locations by the AERONET

(Dubovik et al., 2000; Holben et al., 2001) (http://aeronet.gsfc.nasa.gov) and the SKYNET (Nakajima et al., 1996; 2007) networks, both of which utilize sun-sky scanning radiometer instrumentation. Aerosol absorption retrievals have also been demonstrated by Multifilter Rotating Shadowband Radiometer (MFRSR) instruments (Harrison et al., 1994) at VIS (Kassianov et al., 2005) and UV wavelengths (Bigelow et al., 1998; Petters et al., 2003; Krotkov et al., 2005ab) as well as spectrometers (Harrison et al., 1999; Bais et al., 2005; Barnard et al., 2008). The shadowband technique for aerosol

absorption retrievals does not require separate calibrations for direct and diffuse measurements and allows more frequent (up to one minute) measurements. This technique is more accurate at small solar zenith angles (SZA) (Krotkov et al., 2005ab) complementing AERONET standard almucantar inversions, which are less sensitive for small SZAs (Dubovik et al., 2002).

SKYNET is a ground-based international remote sensing network dedicated for aerosol-cloud-radiation interaction research

(Nakajima et al., 1996; 2007). Using the direct sun and diffuse sky radiance aerosol column average optical properties (e.g., aerosol optical depth (AOD), effective single scattering albedo (SSA), refractive index, and volume particle size distribution (PSD)) are retrieved using standard processing software SKYRAD.pack (Nakajima et al. 1983; 1996). The ability for UV (340 and 380 nm) channels equipped with the PREDE POM-02 sky radiometer used by SKYNET is investigated in this study. Recent comparison studies focused on visible and NIR wavelengths (Che et al., 2008; Estellés et al., 2012; Khatri et

al., 2016) due to the lack of co-incident measurements of aerosol and gaseous absorption properties in the UV. Using SKYNET measurements in Hefei, China, Wang et al. (2014) reported smaller SSA at 380 nm during the autumn and winter (0.91 – 0.93) than that in spring and summer (0.95 – 0.97). They explained lower SSA by combined BC/BrC absorption in smoke from the local farm burning in autumn and from local heating in winter. Their study showed that SSA seasonal variability is smaller than ~0.05. Thus, evaluation and reduction of the uncertainty in the SKYNET SSA retrieval,

particularly at UV wavelengths, is important for aerosol speciation and radiative forcing studies.

This study compares the SKYNET SSA retrievals in extended UV–NIR wavelengths with the SSA derived from a combination of AERONET (Dubovik et al., 2002), MFRSR (Krotkov et al., 2005ab), and Pandora (Herman et al., 2009) inversions (hereafter referred to as AMP) in Seoul, South Korea during and after KORUS-AQ international field campaign



in 2016 (Holben et al., 2017). This study provides first comparisons of the SKYNET and MFRSR SSA retrievals in the UV wavelengths. It also facilitates future comparisons of independent satellite SSA retrievals in the UV from the OMI (Jethva and Torres, 2011; Jethva et al., 2014).

## 2 Experimental site and instrumentation

The data used in this study include measurements from Hampton University's UV- and VIS-MFRSR shadowband radiometers (head number 582 and 579, respectively), a SKYNET sun-sky radiometer (Nakajima et al., 1996; 2007) and an AERONET sun-sky radiometer (Holben et al., 1998) from April to August, 2016 on the roof of the Science Hall, Yonsei University in Seoul, South Korea. Concurrently, an international air quality field study, called the Korea U.S.-Air Quality (KORUS-AQ), has been carried out over the South Korean peninsula from May to June 2016
(https://espo.nasa.gov/home/korus-aq/content/KORUS-AQ). Seoul has high levels of urban pollution since it is a metropolitan region with a population of 25 million, including significant transportation and industrial emissions sources. Also, Seoul is located downwind of regions that include heavy aerosol pollution sources: primarily fossil fuel combustion from industrial and urban areas in Inchon, South Korea and East China, plus biomass burning aerosols from wildfires and crop fires locally and remotely in North Korea, China, Russia, as well as airborne dust from the Taklimakan and Gobi
deserts.

To measure aerosol column optical properties from these sources, the modified UV- and VIS-MFRSR instruments were installed on the roof of the Science Hall, Yonsei University in Seoul, South Korea. The Yankee Environmental Systems (YES) UV- and VIS-MFRSR sensors were modified at the U.S. Department of Agriculture (USDA) UV-B Monitoring and
Research Program (UVMRP) at the Natural Resource Ecology Laboratory, Colorado State University, to facilitate their operation in conjunction with AERONET Cimel sun-photometers. The manufacturer supplied 300-, 317-, and 368-nm UV-MFRSR filters were replaced with 440-, 340-, 380-nm Cimel filters, respectively, used by AERONET. In addition, a 440-nm Cimel filter was added to an unfiltered pyranometer slot of the VIS-MFRSR sensor. Domes were also added to both instruments to prevent Teflon diffuser contamination (Krotkov et al., 2009). These UV and VIS-MFRSR instruments are part
of the USDA UV-B monitoring and Research Program (UVMRP: http://uvb.nrel.colostate.edu/UVB/index.jsf). All MFRSR instruments in the UVMRP network are regularly characterized for their spectral, angular and radiometric responses at the NOAA Central UV Calibration Facility (CUCF: https://www.esrl.noaa.gov/gmd/grad/calfacil/cucfhome.html) in Boulder, Colorado, U.S. The combined set of modified UV- and VIS-MFRSR instruments measures direct solar and diffuse sky irradiances at 13 narrow spectral bands with central wavelengths from the UV to the NIR: 305, 311, 325, 332, 340, 380, 415,
440, 500, 615, 673, 870, and 940 nm. The 440-nm filter common to both MFRSR sensors and to the CIMEL photometer provides spectral overlap between the inversion procedures applied to the three sensors using the procedure described by Krotkov et al. (2005ab) and discussed here in detail. Furthermore, Yonsei University has been operating a CIMEL





sunphotometer as part of the AERONET network as well as a new Pandora spectrometer system to measure trace gases (ozone, $NO_2$, $SO_2$, and HCHO) (Herman et al., 2009). These co-located instruments facilitate the AERONET-to-MFRSR calibration transfer and help in comparing aerosol absorption products such as the imaginary part of the refractive index ($k$), single scattering albedo (SSA), and absorption aerosol optical depth (AAOD). A summary of the instruments can be found in

Table 1.

### 3 Data and Methodology

#### 3.1 MFRSR on-site calibration

Improving the MFRSR observational protocol and daily on-site calibration are critical for accurate measurements of aerosol column absorption. The MFRSR on-site calibration is determined by daily comparisons with the AERONET sun-

photometers, since AERONET measured AOD is highly accurate at ~0.01 to 0.02 with the higher values in the UV (Eck et al., 1999).

We apply corrections for dark current offset, angular response, and instrumental tilt to produce corrected voltages derived from raw voltages measured by MFRSRs. The angular response correction was performed by using the spectral and cosine

response measured at NOAA Central UV Calibration Facility (Krotkov et al., 2005a). To compensate for possible levelling errors, the tilt correction was applied in conjunction with the cosine correction (Alexandrov et al., 2007; Mok, 2017).

We use the estimate of the calibration constant for each individual 1-minute MFRSR measurement at each wavelength (*i.e.*, extraterrestrial voltage, $V_0(\lambda,t)$) calculated using equation (1) to normalize measured direct and diffuse voltages (same

calibration in shadowing technique) and as a quality assurance tool to retain only the best quality measurements consistent with the AERONET AOD measurements. The outlier measurements with $\ln(V_0(\lambda,t))$ exceeding 2 standard deviations from the daily average $<V_0(\lambda, t)>$ are iteratively removed and the daily average $<V_0(\lambda, t)>$ is re-calculated iteratively as described in Krotkov et al. (2005a). Any low-frequency diurnal $V_0(\lambda,t)$ variability indicates possible systematic errors (*e.g.*, not perfect levelling, non-complete shadowing, and/or electronics problems). To reduce systematic errors and outliers, time periods are

selected when $V_0$ does not vary with time (Krotkov et al., 2005a) and only those MFRSR measurements meeting these quality assurance criteria are retained for inversions.

$$\ln V_0(\lambda,t) = \ln(V_{dirn}(\lambda,t)) + \sec(SZA(t)) \left[ \tau_a(\lambda,t) + \tau_R(\lambda,t) + \tau_{NO_2}(\lambda,t) + \tau_{O_3}(\lambda,t) \right], \tag{1}$$

where $V_{dirn}(\lambda,t)$ is the MFRSR-measured direct-normal voltage, $\tau_a(\lambda,t)$ is the spectrally interpolated AERONET level 2 AOD (Eck et al., 1999) applying second order polynomial interpolation/extrapolation least-squares fit in logarithmic space to

the MFRSR wavelengths using the AERONET spectral AOD($\lambda$,t), $\tau_R(\lambda,t)$ is the Rayleigh optical depth inferred from the measured surface pressure, and $\tau_{NO_2}(\lambda,t)$ and $\tau_{O_3}(\lambda,t)$ are $NO_2$ and ozone optical depths, calculated using a Pandora column



NO$_2$ and ozone measurements, interpolated to MFRSR 1-minute measurements (Herman et al., 2009; Tzortziou et al., 2012). For cases when NO$_2$ and O$_3$ values are not available from a Pandora spectrometer, satellite NO$_2$ and ozone measurements from the OMI are used. In polluted urban regions like Seoul, OMI NO$_2$ measurements are typically lower than ground-based retrievals (Irie et al., 2009; 2012; Ialongo et al., 2016; Krotkov et al., 2017).

Using only the best quality MFRSR measurements, the mean $V_0$ value for a given day ($<V_0(\lambda, t)>$) is calculated and then MFRSR values ($\tau_{a(MFRSR)}(\lambda,t)$) are calculated by inverting equation (1):

$$\tau_{a(MFRSR)}(\lambda,t) = \cos\big(SZA(t)\big) \ln\big(\langle V_0(\lambda)\rangle / V_{dirn}(\lambda,t)\big) - \tau_R(\lambda,t) - \tau_{NO_2}(\lambda,t) - \tau_{O_3}(\lambda,t) \,, \qquad (2)$$

Finally, the measurements are only used when the root-mean-squared (RMS) of ($\tau_{a(MFRSR)}(\lambda,t) - \tau_{a(AERONET)}(\lambda,t)$) < 0.01, where $\tau_{a(AERONET)}(\lambda,t)$ is the spectrally interpolated AERONET level 2 AOD applying second order polynomial interpolation/extrapolation least-squares fit in logarithmic space (Eck et al., 1999) to the MFRSR central band wavelengths. The spectral interpolation error is typically less than 0.01. In MFRSR SSA retrievals, we use $\tau_{a(AERONET)}(\lambda,t)$.

**3.2 MFRSR inversion technique**

Currently ground measurements of column effective refractive index and single scattering albedo (SSA) are limited to the 4 discrete visible and near infrared wavelength bands by AERONET almucantar inversions (440, 675, 870, and 1020 nm). An AERONET CIMEL sunphotometer has 340 and 380 channels but does not provide SSA inversions. However, sky radiance measurements are currently made by many instruments at 380 nm and the SSA at 380 nm will be a future data product. To
extend SSA retrievals into UV and other wavelengths (Table 1), our method combines synchronous co-located measurements by AERONET, MFRSR, and Pandora ensuring consistent retrievals of AOD, particle size distribution (PSD), real part of the refractive index ($n$), and gaseous absorption (*e.g.*, by ozone and NO$_2$). We also use consistent spectral surface albedo (monthly climatological values) derived from satellite MODIS surface albedo data (Moody et al., 2005; Eck et al. 2008). MFRSR-measured Diffuse/Direct (DD) irradiance ratios are fitted with a forward radiative transfer model coupled
with the Mie scattering code (Arizona code (Herman et al., 1975)) to estimate only one forward model parameter: column effective imaginary part of refractive index ($k$) independently for each MFRSR spectral channel (Krotkov et al., 2005b).

The procedure of the combined AMP retrievals is summarized as a flowchart (Figure 1). Required ancillary input parameters such as PSD, $n$, surface pressure, and surface albedo are taken from co-located near simultaneous AERONET inversions
(Dubovik et al., 2002). Gaseous absorption of column ozone and NO$_2$ are accounted for using ground-based direct-sun retrievals by Pandora spectrometers (Herman et al., 2009; Tzortziou et al., 2012) or satellite data from Aura/OMI overpass when Pandora data are not available. AOD is obtained either from MFRSR inferred direct (total - diffuse) irradiances





(corrected for laboratory measured angular response) or AERONET direct sun measurements. Then, the Mie-RT model is iterated to find the $k$ value, which minimizes the difference between calculated and measured diffuse to direct (DD) irradiance ratio. The fitted $k$ value together with AERONET inferred PSD and $n$ at 440 nm is converted to SSA using Mie calculations assuming spherical particles (Krotkov et al., 2005b). As shown in Figure 2, the Angstrom Exponent (AE)
observations from AERONET are mostly higher than unity, which is typical for predominantly fine mode pollution aerosols.

We estimate retrieval errors of $k$ ($\Delta k$) and SSA ($\Delta \omega$) using combined Mie-RT code to calculate the finite difference normalized Jacobians (J):

$$J_{k,DD} = \frac{\frac{\Delta k}{k}}{\frac{\Delta DD}{DD}} \, , \tag{3}$$

$$\Delta k = J_{k,DD} \frac{\Delta DD}{DD} k \, , \tag{4}$$

$$J_{\omega,k} = \frac{\frac{\Delta \omega}{\omega}}{\frac{\Delta k}{k}} \, , \tag{5}$$

$$J_{\omega,DD} = J_{\omega,k} J_{k,DD} \, , \tag{6}$$

$$\Delta \omega = J_{\omega,DD} \frac{\Delta DD}{DD} \omega \, , \tag{7}$$

Using equation (7), the error of SSA ($\Delta \omega$) is calculated as shown the vertical bar in Figure 3b and 3c. Assuming constant 3%
accuracy in the measured DD ratio (Equation (3) – (4)) the calculated SSA retrieval error $\Delta \omega$ is inversely proportional to AOD, but typically is less than 0.02 for AOD at 440 nm, $AOD_{440} \geq 0.2$.

### 3.3 Sky radiometer (SKYNET)

In analysing SKYNET sky radiometer measurements conducted here, we use the Sky Radiometer analysis package from the
Center for Environmental Remote Sensing (SR-CEReS) version 1. As the main program, skyrad.pack version 5 (Hashimoto et al., 2012) is implemented to retrieve aerosol properties in SR-CEReS along with all pre- and post-processing programs for the purpose of the near-real time data delivery. Two kinds of calibration approaches were considered for the present study. The first approach is to use the static calibration constants. We derived the static calibration constants through the comparison with the reference sky radiometer, which was calibrated at the Mauna Loa Observatory (MLO) in December
2015, and through the direct calibration at the MLO in October and November 2016. The second approach is to use dynamic on-site calibration method, based on the idea of the Improved Langley method (Campanelli et al., 2007; Khatri et al., 2016). Since the first method is not able to account for the possible temperature variations on a daily to monthly time scale during very hot summer for instance, the latter calibration method was selected in this study to estimate the daily calibration





constant ($\langle F_0 \rangle$). To minimize the temporal stability of $\langle F_0 \rangle$ to $\pm 1 - 3\%$ due to temperature variation and consider the consistency with the above-mentioned static calibration constants, the following method was used in this study.

$F_0$ was first calculated tentatively for each measurement, where aerosol parameters were retrieved utilizing ratios of aureole
radiance to direct radiance assuming the known field of view of the instrument (Tanaka et al., 1986; Nakajima et al., 1996):

$$F = \frac{F_0}{R^2} \exp(-m\tau) ,$$  (8)

where $F$, $m$, $\tau$, and $R$ are the measured intensity, the air mass, total (Rayleigh + aerosol + ozone) optical depth, and the Sun-Earth distance, respectively, and all are given quantities.

However, the uncertainties arise because 1) $\tau$ has uncertainty in the absorption component and 2) $m$ has uncertainty due to
the refraction at high SZAs (corresponding to high $m$ values). To estimate $\langle F_0 \rangle$, we use the statistical approach as follows: 1) two-month period ($\pm 30$ days of the target day) is used to calculate statistics, 2) only clear sky $F_0$ values obtained within the lowest 1/3 of all $m\tau$ values are used, and 3) only $F_0$ values within their 3 standard deviations are used. Regarding the threshold of 1/3, we tested other thresholds and found that the choice is not critical. This threshold was likely best to keep the sufficient number of data to determine $\langle F_0 \rangle$ at small $m\tau$ values. Then, the average of those data is regarded as the final $\langle F_0 \rangle$
value for the target day. This statistical approach is taken as a pre-processing and different from previous studies. While daily $\langle F_0 \rangle$ values for entire UV-VIS-NIR channels have not been given in previous studies, reanalysis of their observation data by this approach is preferable to confirm the consistency. For the cloud screening, this study uses the method of Khatri and Takamura (2009) but not including global irradiance data from a pyranometer.

## 4 Results and discussion

### 4.1 Comparison of single scattering albedo between AERONET and MFRSRs

First, the individual 1-minute UV- and VIS-MFRSR retrieved SSA at 440 nm, $SSA_{440}$, are compared to demonstrate the high degree of consistency for a combined set of modified UV- and VIS-MFRSR instruments (Figure 3a). The correlation coefficient between UV-MFRSR and VIS-MFRSR retrieved $SSA_{440}$ is 0.98, the estimated standard deviation of MFRSR $SSA_{440}$ uncertainty (standard MFRSR uncertainty (Fioletov et al., 2016)) is ~0.007, and the mean $SSA_{440}$ difference (bias) is
less than 0.002. Next, $SSA_{440}$ from AERONET level 1.5 inversions are compared with the ~32-minute average $SSA_{440}$ retrievals from either the UV-MFRSR (Figures 3b) or VIS-MFRSR (Figure 3c). For the time averaging interval we use $\pm 16$ minutes from the AERONET inversion time. Both instruments provide the best quality SSA retrievals at high turbidity conditions ($AOD_{440} \geq 0.4$) (Dubovik et al., 2002; Krotkov et al., 2005b; Mok et al., 2016). For these conditions, the average $SSA_{440}$ from either UV-MFRSR or VIS-MFRSR (~0.92) are in excellent agreement with the corresponding AERONET
average $SSA_{440}$ (~0.93), less than 0.01.





Relaxing the AERONET level 2 inversion $AOD_{440} \geq 0.4$ criterion allows for analysing a larger statistical sample of the MFRSR-AERONET matchups (Figure 3). However, the mean $SSA_{440}$ values using relaxed AOD filter ($AOD_{440} \geq 0.2$, shown as blue and red dots) are reduced by ~0.02 compared to the restricted sample using AERONET level 2 criteria

($AOD_{440} \geq 0.4$, shown as red dots). The SSA variability (standard deviation) using the relaxed filter is insignificantly increased (less than 0.01) compared to using the restricted filter. The increased variability reflects cases with smaller AOD, showing stronger absorption (SSA ~ 0.9). The root mean square deviation (RMSD) is higher for lower AOD cases (~ 0.030 – 0.034 for $0.2 \leq AOD_{440} < 0.4$) than for higher AOD cases (~0.022 for $AOD_{440} \geq 0.4$) (Table 2) as shown in previous studies (Dubovik et al., 2002; Estellés et al., 2012). The good agreement in SSA at the common overlapping wavelength 440 nm

from UV-MFRSR, VIS-MFRSR, and AERONET level 1.5 provide additional justification to using the MFRSR and AERONET level 1.5 inversions with $AOD_{440} \geq 0.2$. Thus, we utilize the combined AMP SSA retrievals for $AOD_{440} \geq 0.2$ to compare with the SKYNET SSA retrievals.

Comparing low scatter in $SSA_{440}$ differences between UV-MFRSR and VIS-MFRSR (Figure 3a), Figures 3b and 3c show

larger scatter between either UV-MFRSR (Figure 3b) or VIS-MFRSR (Figure 3c) and AERONET $SSA_{440}$. We explain this by several possible reasons. The two MFRSR instruments measure the total sky hemispherical irradiance affected by even small cloud fraction, whereas AERONET has the ability to filter out scattered cumulus from the symmetry check done on directional sky radiances in the almucantar scan. Therefore, it is possible that some MFRSR SSA retrievals are more affected by the presence of scattered clouds than the AERONET retrievals. Another potential source of scatter between AERONET

and MFRSR $SSA_{440}$ retrievals could be gaseous absorption by $NO_2$ that is not completely accounted for in the AERONET retrievals in Version 2. Next, coarse mode fraction, which varies approximately from ~5% to 50% in South Korea for these paired measurements (Figure 2), primarily by the mixture of dust and urban aerosols, could affect the MFRSR retrievals which assume spherical particles, while dust is complex in shape. Additionally, coarse mode size particles scatter much more strongly in the forward direction than fine mode particles, thereby resulting in additional variable uncertainty in the solar

aureole corrections made to account for the sky fraction blocked by the shading band in the MFRSR instrument (di Sarra et al., 2015).

Figures 4a and 4b compare AERONET and MFRSR SSA at longer NIR wavelengths: 673 and 870 nm ($SSA_{673}$ and $SSA_{870}$), respectively. Note that the average AOD at 675 and 870 nm (0.34 and 0.24, respectively) are lower than the $AOD_{440}$ ~0.6, as

the average Angstrom Exponent (440 – 870 nm) is 1.30 (Figure 2). The lower AOD at 675 and 870 nm is main reason for the larger SSA retrieval noise (RMSD = 0.025 and 0.026 for $AOD_{440} \geq 0.4$). However, the discrepancies between mean AERONET SSA and mean MFRSR SSA at 675 and 870 nm are less than 0.02 regardless of whether the relaxed or strict filter is adopted. The MFRSR calculated SSA uncertainties are less than ~0.03, which is typical AERONET SSA retrieval uncertainty. Such agreement allows us to compare the AMP SSA with the SKYNET SSA as discussed below.




**4.2 Comparison of single scattering albedo between AMP and SKYNET**

Previous comparison studies of retrieved aerosol optical properties between AERONET and SKYNET (Che et al., 2008; Estellés et al., 2012) show typically good agreement for AOD. However, Khatri et al. (2016) found that the SKYNET SSA

was overestimated compared to AERONET SSA inversions at visible and NIR wavelengths. None of previous studies (Che et al., 2008; Hashimoto et al., 2012; Khatri et al., 2016) performed the intercomparison of SSA in the UV wavelengths. This study is the first to compare SKYNET SSA retrievals at UV to NIR wavelengths using co-located near simultaneous (±16 minutes) AMP retrievals in Seoul in 2016. Figure 5 shows SSA comparison results between AMP and SKYNET in extended wavelength range from 340 to 870 nm. Correlation between the two SSA retrievals is moderately high, decreasing at 675 and

870 nm due to higher uncertainty in the SSA retrievals at lower AOD. The SSA scatter could result from small AOD differences, which are independently measured in SKYNET and AMP retrievals. Nevertheless, the mean absolute SSA differences are less than 0.02, within uncertainties in the SSA retrievals. We found that, on average, the SKYNET SSA at UV wavelengths is lower compared to the AMP SSA (Figure 5). The likely source of the bias could be the spectrally invariant surface albedo (0.1, Figure 6) assumed in SKYNET SSA retrievals. This incorrect assumption leads to the

underestimated SSA values in UV, even if AOD retrievals are accurate (Hashimoto et al., 2012).

**4.3 Main factors of discrepancy**

Previous studies (Che et al., 2008; Hashimoto et al., 2012; Khatri et al., 2016) show that the SKYNET SSA is overestimated compared to the AERONET SSA in the visible and NIR wavelengths. Khatri et al. (2016) reported that the main cause of

these discrepancies is systematic difference in absolute calibration of sky radiances. Specifically, the overestimation of sky radiances in SKYNET measurements results in the SSA overestimation as compared to AERONET. Unlike previous studies we found that the SKYNET SSA is in good agreement with the AMP SSA in the visible wavelengths, but lower in the UV wavelengths (Figure 5 and Table 3).

**4.3.1 Surface albedo**

Surface albedo has an important impact on the retrievals of SSA in the UV region (Corr et al., 2009). The AMP inversions use the AERONET-provided spectral surface albedos at 440, 670, and 870 nm derived from MODIS surface BRDF/albedo product (Moody et al., 2008). The shortest wavelength at which surface albedo is available is 440 nm. Therefore, we assumed that the surface albedo at 440 nm applies to MFRSR retrievals in all the UV wavelengths.





Figure 6 compares surface albedo used in AMP inversions with that assumed in SKYNET inversions. There is little variability in MODIS-derived climatological surface albedo (Moody et al., 2008) assumed in AERONET inversions (±0.01) at 440 nm. The SKYNET retrievals compared here use the spectrally invariant surface albedo (0.1) at all wavelengths. The spectrally independent SKYNET-assumed surface albedo 0.1 is close to the AERONET surface albedo at 675 nm (Figure 6).

However, it greatly deviates from the MODIS surface albedo at 440 nm and 870 nm (~0.04 and ~0.2, respectively used by AERONET and AMP retrievals). The overestimated value of surface albedo in the SKYNET inversions will lead to an underestimated value of SSA at near UV wavelengths: 340, 380, and 400 nm (Hashimoto et al., 2012). As seen in Figure 5, this explains the lower SKYNET SSA compared to AMP retrievals.

Re-processing the SKYNET inversions using spectrally varying surface albedo (Figure 6), consistent with the AERONET retrievals significantly improves agreement between the SKYNET SSA and the AMP SSA (Figure 7 and Table 3). The updated surface albedo in the SKYNET inversions significantly increases the SSA (by ~0.01) at wavelengths from 340 to 500 nm. The mean SSA differences between AMP and re-processed SKYNET are reduced to ~0.013, 0.002, and 0.003 (for $AOD_{440} \geq 0.4$) at 340, 380, and 400 nm, respectively. The root-mean-squared differences are also reduced (RMSD < 0.02) at

these wavelengths (Table 3). Thus, using consistent surface albedo significantly reduces systematic biases between SKYNET, MFRSR (AMP) and AERONET retrievals, particularly at UV wavelengths.

### 4.3.2 AOD

The close agreement of AOD (i.e., better than 0.01) is a critical pre-condition for SSA comparison, since the overestimation

in AOD leads to the underestimation in SSA and vice versa (Dubovik et al., 2000; Khatri et al., 2016). The discrepancy of AOD is typically attributed to problems in instrumental calibrations (Khatri et al., 2016). Figure 8 shows the only significant AOD differences between AMP and SKYNET at a wavelength of 340 nm, where the mean bias difference (MBD) and RMSD were ~ 0.030 and ~ 0.044, respectively. The differences of mean AOD were less than ~0.01 at all other wavelengths. We conclude that AOD differences were not significant in our SSA comparisons at wavelengths longer than 340 nm.

### 4.3.3 Atmospheric gas absorption

The AMP inversions account for effects of gaseous (ozone and $NO_2$) absorption in the UV and visible wavelengths. However, the gaseous absorption (ozone and $NO_2$) is not taken into account in the sky radiances that are inverted in the AERONET Version 2 retrievals. In the SKYNET retrievals, only fixed column ozone (300 DU) is considered without the

$NO_2$ absorption. In the upcoming AERONET Version 3 data base, the ozone and $NO_2$ absorption will be accounted for in sky radiances by using monthly climatological values from Aura/OMI satellite retrievals (Bhartia, 2005; Krotkov et al.,



2017). However, errors in the daily SSA retrievals could be introduced if one uses a fixed climatological value of column $NO_2$ (Corr et al., 2009) at UV and blue wavelengths.

As discussed in section 4.3.1, the agreement between the AMP and SKYNET SSA is improved by using consistent MODIS-
derived surface albedo (0.04) in the SKYNET SSA retrievals at 340, 380, and 400 nm. Still, the SKYNET-derived SSA (for $AOD_{440} \geq 0.4$) shows a slight underestimation compared to the AMP-derived SSA at these wavelengths. To investigate $NO_2$ gaseous absorption as possible cause, we modified our AMP SSA inversion assuming zero $NO_2$ absorption and found SSA decreased by ~0.004 – 0.007 at 340, 380, and 415 nm, closer with SKYNET retrievals. Thus, accounting for $NO_2$ absorption should further reduce the negative bias in SKYNET SSA retrievals. The $NO_2$ effect on SSA retrieval is largest for small
AOD and could lead to incorrect interpretation of aerosol composition (Krotkov et al., 2005c). We also found that including $SO_2$ absorption (average $SO_2$ column amount in Seoul is < 1 Dobson Unit, $1DU = 2.69*10^{16}$ molecules $cm^{-2}$) (Krotkov et al., 2016) results in negligible increases in SSA (~0.003 at 305 nm and less at longer wavelengths).

**4.4 SSA spectral dependence**

As shown in Figure 9, AMP and SKYNET SSA retrievals using the AERONET spectrally varying surface albedo are in good agreement at all wavelengths. The SSA typically decreases with wavelength in the visible and NIR wavelengths, reaches flat maximum between 415 – 500 nm and decreases sharply in shorter UV wavelengths. This can be explained by the mixture of spectrally flat absorbing black carbon and selectively UV-absorbing aerosols (i.e., brown carbon, dust). The detailed investigation relating aerosol type and SSA spectral dependence will be discussed in future studies. Here we
conclude that AMP and SKYNET retrievals are in good agreement, both allowing for measuring aerosol absorption and its spectral dependence.

**5 Summary and Conclusion**

This study uses simultaneous measurements from co-located AERONET, MFRSR, and Pandora instruments to ensure accurate measurement of aerosol extinction optical depth, in order to provide consistent inversions of aerosol column absorption properties between UV and visible wavelengths, and partition between aerosol and gases absorption. Using this technique, we retrieved the column spectral SSA in the UV, visible, and NIR wavelength and performed the SSA comparisons between AERONET and MFRSR retrievals. The SSA comparisons between AERONET and MFRSR are in
good agreement, showing the mean SSA difference is less than 0.01 at common wavelength 440 nm for both conditions of $AOD_{440} \geq 0.4$ and $AOD_{440} \geq 0.2$. The latter condition, called the relaxed filter, increases the number of AERONET-MFRSR

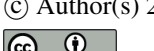



matchup by a factor of ~1.5 and is used for comparisons with SKYNET. As a result, our approach can provide SSA at wavelengths AERONET cannot provide and can be compared with the SKYNET SSA.

The new finding is the underestimation of the SKYNET SSA in the UV which has not been previously discussed. The
5   underestimation could be explained, in part, by the use of the unrealistically high surface albedo (0.1). The UV surface albedo should not be larger than the MODIS derived values at 470 nm (~0.04), used in AERONET SSA retrievals at 440 nm. Following this recommendation, updating the surface albedo in the SKYNET inversions to the average AERONET value of ~0.04 significantly reduces average differences in SSA (~0.01) in the near UV.

10   This study demonstrates the consistency of the column aerosol spectral absorption derived from the AMP and SKYNET inversions in the extended wavelength region. Specifically in UV wavelengths this study presents the first comparison of the column average SSA measured by independent ground-based techniques. It is found that SKYNET provides more reliable SSA at UV wavelengths (340 and 380 nm) on the condition that the spectrally varying surface albedo and $NO_2$ absorption are accounted for. Considering the results of this study, the SSA measurements presented here are more essential to answer
15   how the UV light absorbing aerosols affect air quality, surface UV radiation, and tropospheric oxidation capacity, which remains highly uncertain. In addition, retrieved aerosol absorption in the UV contributes to improving the classification algorithm of the columnar aerosol types (Kim et al., 2007; Choi et al., 2016; Mok et al., 2016) and validating satellite SSA retrievals from the current (Aura OMI (Jethva and Torres, 2011) and SNPP OMPS) and future satellite atmospheric composition missions (TROPOMI, TEMPO, GEMS, and Sentinel-4).





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

**Acknowledgements**

J. Mok was supported by the University of Maryland/ESSIC–NASA cooperative agreement. J. Kim, J.-H Koo, and S. Go were supported by the Korea Ministry of Environment (MOE) under grant: "Public Technology Program based on
10   Environmental Policy (2017000160001)". The authors acknowledge support from NASA Earth Science Division, Radiation Sciences and Atmospheric Composition programs and the AERONET project at GSFC. The authors also thank the AERONET team, C. M. Wilson from the NOAA Central UV Calibration Facility (CUCF), plus G. Janson, and B. Olson, from the USDA UV-B monitoring and Research Program (UVMRP).



**Table 1. Instruments and wavelengths of retrieved absorption properties.**

| Instruments | Measurements | Wavelengths (nm) |
|---|---|---|
| CIMEL sun and sky photometers (AERONET) | Direct sun and almucantar sky radiance, Filters (2 – 10 nm) | 440, 675, 870, 1020 |
| Modified UV-MFRSR (#582) | Diffuse and total irradiance, Filters (2 nm) | 305, 311, 325, 332, 340, 380, 440 |
| Modified VIS-MFRSR (#579) | Diffuse and total irradiance, Filters (2 nm) | 415, 440, 500, 615, 673, 870, 940 |
| Sky radiometer (SKYNET) | Sun and sky radiance, Filters (10 nm) | 340, 380, 400, 500, 675, 870, 1020 |

**Table 2. Comparison of SSA at 440 nm between AERONET and AMP inversions via UV-MFRSR and VIS-MFRSR.**

| | $0.2 \leq AOD_{440} < 0.4$ | | $AOD_{440} \geq 0.4$ | |
|---|---|---|---|---|
| | AERONET | MFRSR | AERONET | MFRSR |
| AERONET and UV-MFRSR matchup | | | | |
| Mean | 0.892 | 0.869 | 0.929 | 0.918 |
| Standard deviation | 0.038 | 0.038 | 0.038 | 0.042 |
| Correlation | 0.77 | | 0.89 | |
| Number | 24 | | 45 | |
| RMSD | 0.034 | | 0.022 | |
| AERONET and VIS-MFRSR matchup | | | | |
| Mean | 0.897 | 0.878 | 0.933 | 0.922 |
| Standard deviation | 0.039 | 0.039 | 0.037 | 0.041 |
| Correlation | 0.82 | | 0.89 | |
| Number | 30 | | 50 | |
| RMSD | 0.030 | | 0.022 | |





**Table 3. Statistical differences between AMP and SKYNET retrieved SSA with spectrally invariant surface albedo=0.01 (in parenthesis) and spectrally varying surface albedo (Figure 6). Statistics, such as root mean square deviation (RMSD), mean difference (MBD), standard deviation (STD), and 95 percentile (U95) of the differences are computed for $AOD_{440} \geq 0.4$ consistent with the quality assured level 2 AERONET inversion data.**

| Wavelength (nm) | RMSD | MBD (AMP-SKYNET) | STD | U95 | Number |
|---|---|---|---|---|---|
| 340 | 0.0172 (0.0249) | 0.0127 (0.0217) | 0.0120 (0.0126) | 0.0363 (0.0495) | 20 |
| 380 | 0.0147 (0.0182) | 0.0020 (0.0111) | 0.0149 (0.0149) | 0.0283 (0.0398) | 20 |
| 400 | 0.0163 (0.0202) | 0.0034 (0.0125) | 0.0164 (0.0163) | 0.0417 (0.0527) | 19 |
| 500 | 0.0255 (0.0241) | -0.0070 (0.0031) | 0.0251 (0.0245) | 0.0461 (0.0587) | 19 |
| 675 | 0.0371 | -0.0017 | 0.0381 | 0.0700 | 19 |
| 870 | 0.0471 (0.0481) | -0.0049 (-0.0004) | 0.0482 (0.0495) | 0.0719 (0.0799) | 18 |





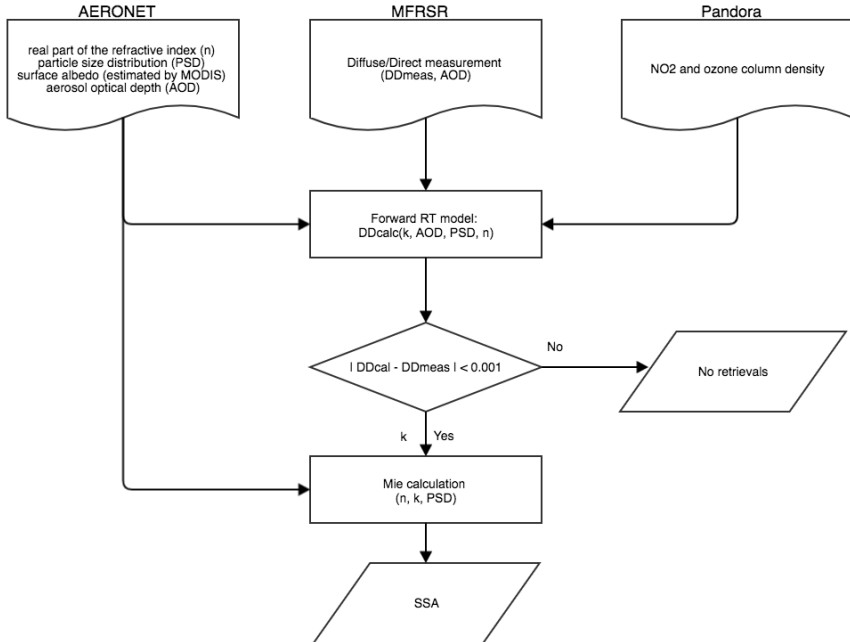

**Figure 1. Flowchart showing the combined AERONET-MFRSR-Pandora (AMP) SSA inversion methodology.**

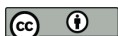



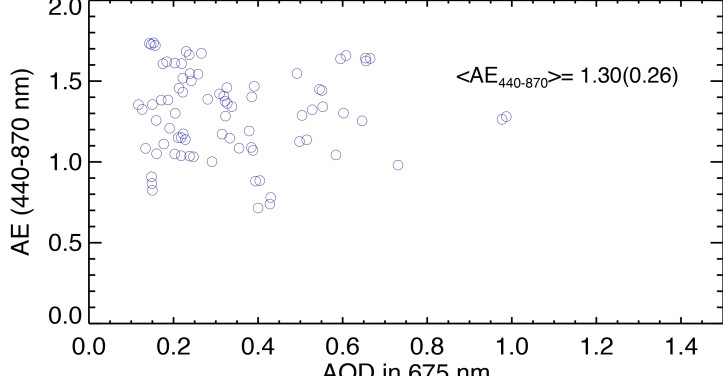

**Figure 2. The Angstrom Exponent (AE) (440 – 870 nm) as a function of AOD at 675 nm. The prevailing values of AE greater than unity characterize the relative influence of fine mode particles during April to August in 2016. The average Angstrom Exponent (440 – 870 nm) is 1.3 and its standard deviation is 0.26.**



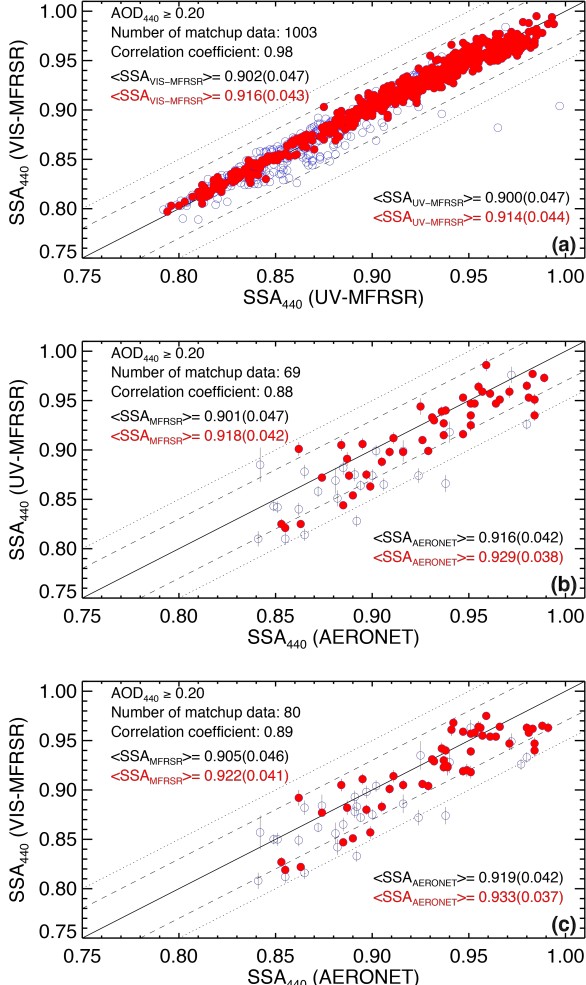

**Figure 3. Comparison between SSA at 440 nm retrieved from AERONET-only and AMP retrievals in Seoul: (a) all 1-minute UV-MFRSR versus VIS-MFRSR retrievals, (b) AERONET inversions versus 32-minute average UV-MFRSR retrievals, and (c) AERONET inversions versus 32-minute average VIS-MFRSR retrievals. MFRSR SSA mean errors are shown assuming 3% error in diffuse to direct ratio. The UV- and VIS-MFRSR SSA in (b) and (c) are averaged within ±16 minutes from the AERONET retrieval time. The dashed lines show SSA agreement within ±0.03, which is assumed SSA error. The dotted lines are ± 0.05 of the 1:1 line. Red color shows comparisons for $AOD_{440} \geq 0.4$, consistent with the best quality Level 2 AERONET inversions. Blue dots indicate retrievals for $0.2 \leq AOD < 0.4$. Combined SSA statistics for $AOD \geq 0.2$ are shown in black. Standard deviation of SSA is indicated in parentheses.**

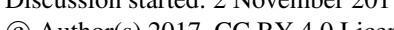



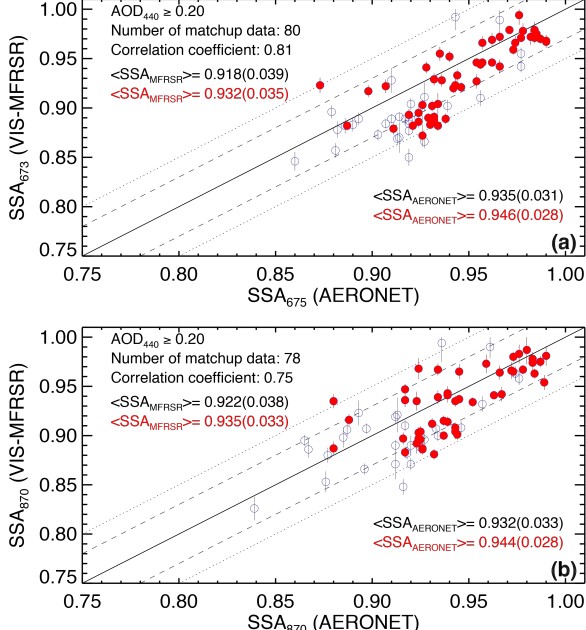

**Figure 4.** Comparison between SSA at 675 and 870 nm retrieved from AERONET almucantar retrievals and SSA at 673 and 870 nm retrieved from MFRSR DD retrievals.





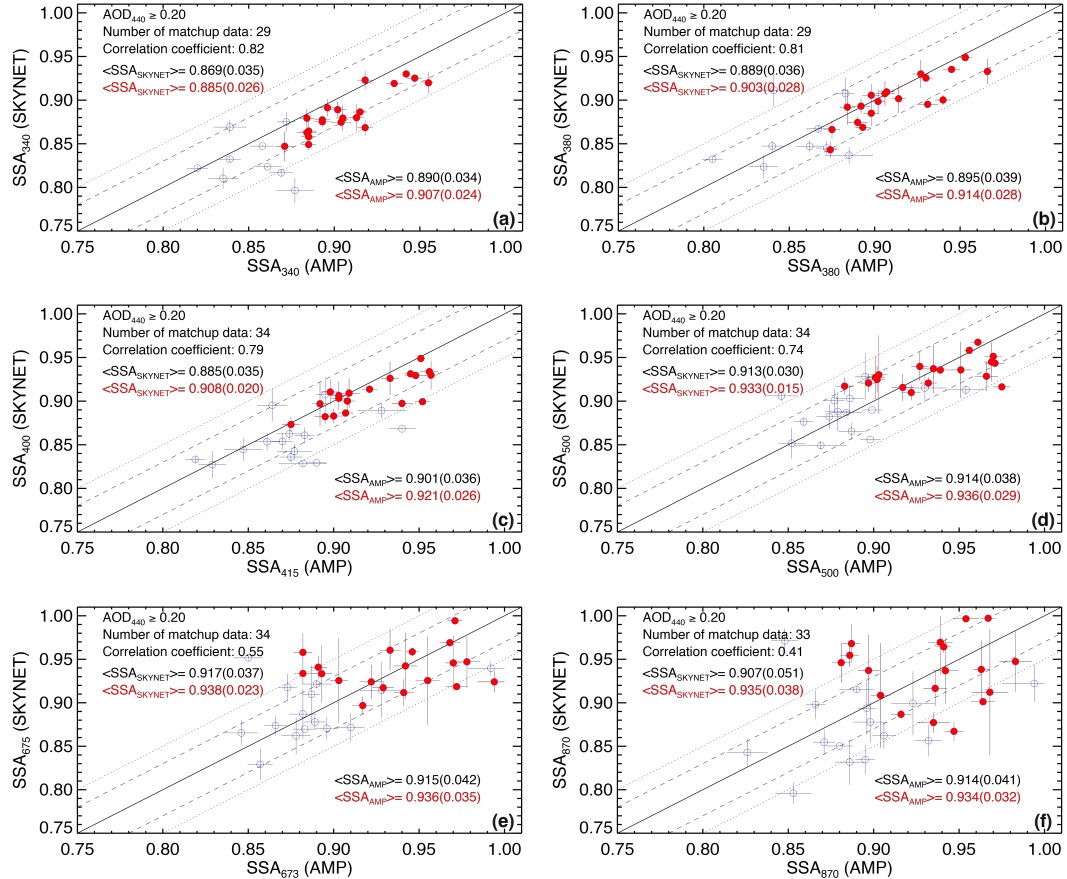

**Figure 5. Comparisons of AMP-retrieved SSA with SKYNET-retrieved SSA using spectrally flat surface albedo (0.1) at all wavelengths. Red dots are filtered using AOD$_{440}$ ≥ 0.4 to correspond the best quality level 2 AERONET data.**





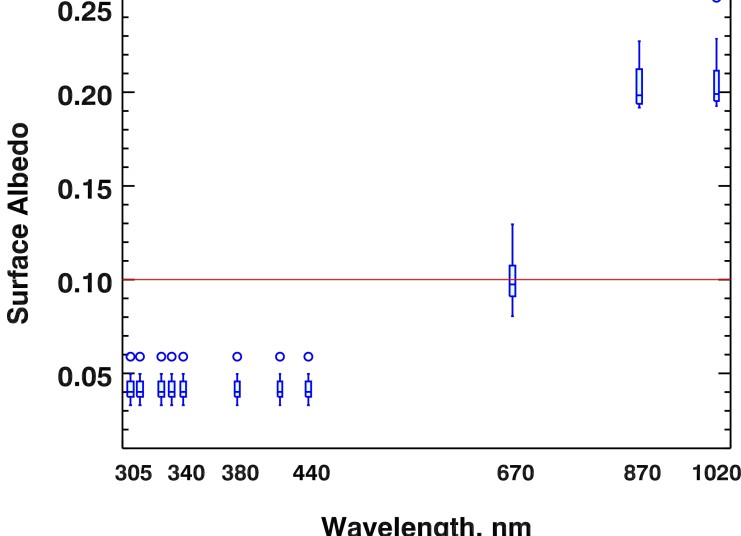

**Figure 6. Surface albedo used for AMP (blue symbols) and SKYNET (red line) SSA inversions. The bottom and top edges of the boxes are located at the sample 25th and 75th percentiles; the whiskers extend to the minimal and maximal values within 1.5 interquartile range (IQR). The outliers are shown in circles. Constant surface albedo of 0.1 assumed for all wavelengths in SKYNET retrievals, is shown as red solid line.**

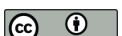



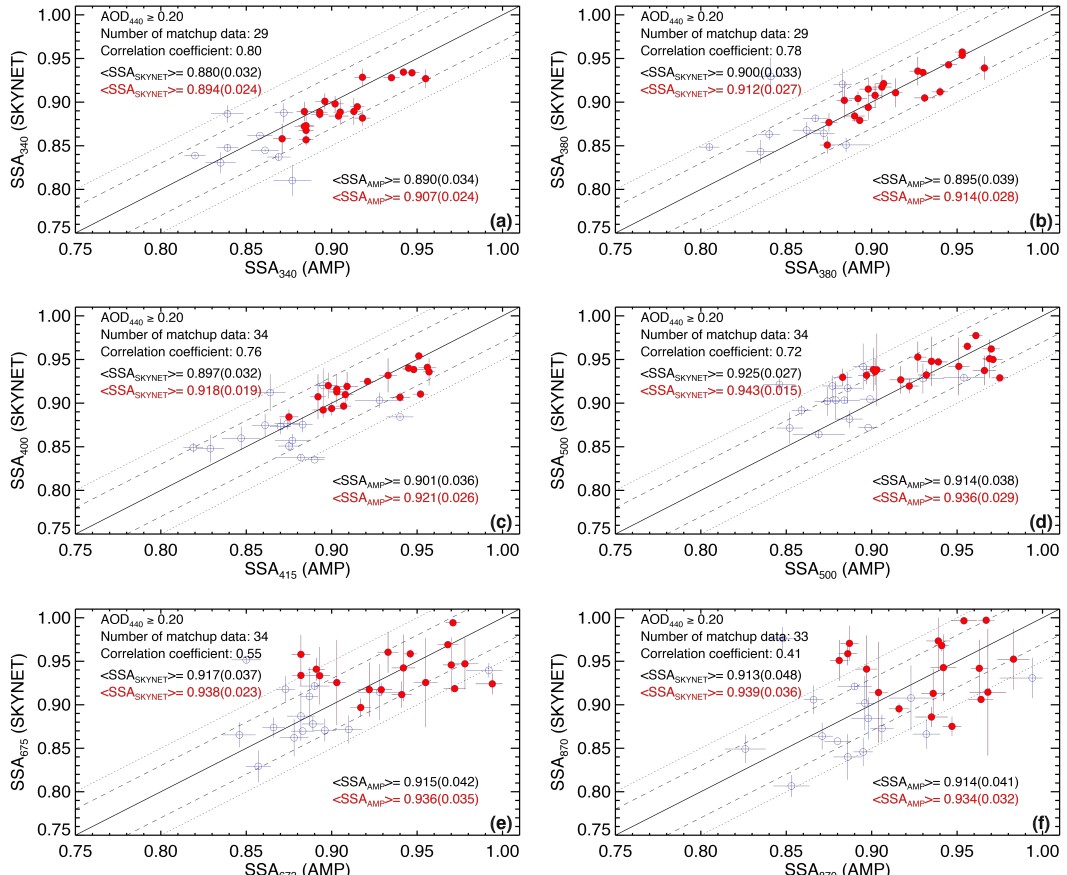

**Figure 7. Re-processed SKYNET SSA at 340, 380, 400, 500, and 870 nm using spectrally varying surface albedo, which corresponds the MODIS-derived surface albedo shown in Figure 6.**





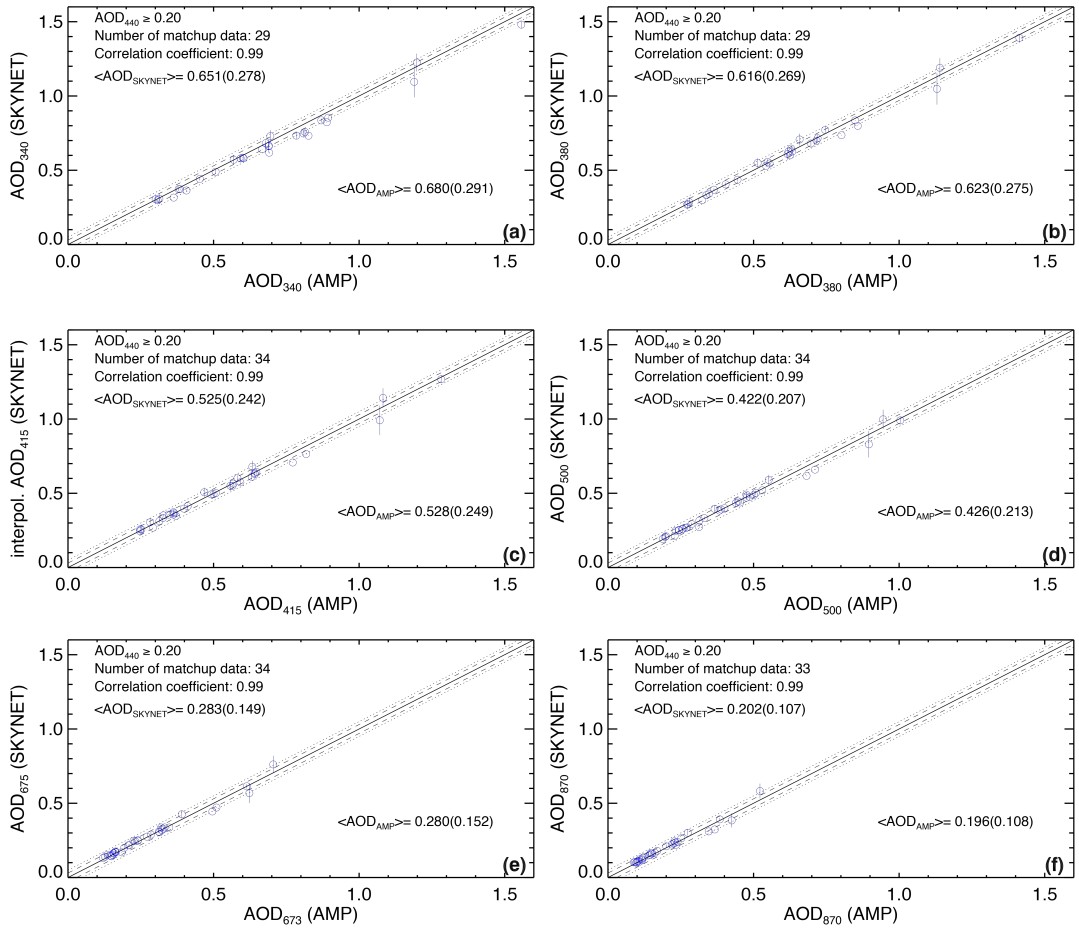

**Figure 8. Comparisons of AMP-retrieved AOD with SKYNET-retrieved SSA using SKYNET retrievals with spectrally varying surface albedo. AMP/AOD is AERONET/AOD used for inversions and/or interpolated to UV wavelengths and times. Dotted and dashed lines are 0.03 and 0.05 offset, respectively.**




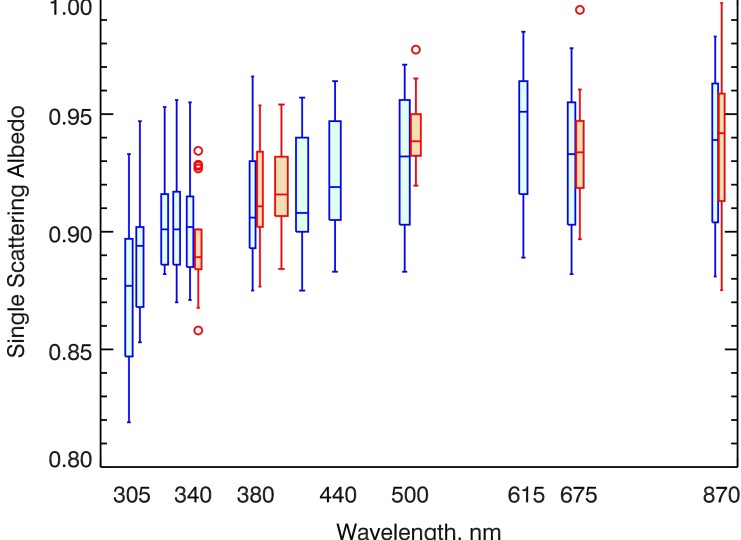

**Figure 9. Combined spectral SSA from AMP-retrievals (blue symbols) and SKYNET retrievals (orange symbols) using MODIS-derived surface albedo shown in Figure 6. The bottom and top edges of the boxes are located at the sample 25th and 75th percentiles; the whiskers extend to the minimal and maximal values within 1.5 IQR. The outliers are shown in circles. The center horizontal lines are drawn at the median values. The whisker-boxes are computed using $AOD_{440} \geq 0.4$ criteria to correspond the best quality level 2 AERONET data.**