# Peer review of "Comparisons of spectral aerosol single scattering albedo in Seoul, South Korea"

_Atmospheric Measurement Techniques, 2017_

## Referee Comment (RC1) · Anonymous Referee #1 · 30 Nov 2017

This manuscript by Mok et al., "Comparisons of spectral aerosol absorption in Seoul, South Korea", presents a comparison of SKYNET-retrieved SSA in the UV with the SSA derived from a combination of AERONET, MFRSR, and Pandora retrievals in Seoul, South Korea in spring and summer of 2016. There have been only a limited number of measurements / measurement campaigns focusing on absorption at UV wavelengths, therefore, the topic of this study is of great interest and relevance. The scope of the paper is both concise and specific, and my minor comments are mainly related to the need to clarify some of the issues. Before publication, the following points should be addressed:

GENERAL COMMENTS:

I was missing some more information and details about the measurements, for instance

regarding the following two points.

1) Figure 5 (and 7) shows a large variability for each SKYNET mean value, so apparently it is shown based on several measurements within 32 minutes, but what is the temporal resolution of SKYNET measurements? I think this was never mentioned.

2) Page 9, Line 25, here solar aureole corrections are mentioned. Please include few sentences to explain this correction in some detail. Also, this same issue applies also, to some extent, to Cimel measurements (diffuse light in FOV). Could you discuss the relative importance of this kind of uncertainty in both measurements?

The focus of the paper is on UV wavelengths and thus one should not perhaps concentrate too much on the longer wavelengths, however I cannot help wondering about the comparison in the Figure 5 and at the wavelengths > 400nm. Now the explanation was given that the larger scatter is due to the lower AOD and related larger uncertainty. However, it is not only the larger scatter, but also the systematic behavior that stands out, e.g. at 500 nm for largest AOD (red points) SKYNET is showing very little SSA variability. SKYNET value is close to 0.93, when AERONET SSA varies from 0.88 to 0.98. AOD is not very low in these cases at 500nm, when it is larger than 0.4 at 440nm. Is there any idea why this happens? Similar pattern and poor agreement seems to be true also at 675nm. Also, the vertical error bars are sometimes strikingly large. In your Figure 8 you show that AODs match well, so what could be the main reason to cause a variability this large SSA variability within a short period of measurements?

Page 9, Lines 14-26. Here are several possible explanations given for a larger scatter between AERONET and MFRSR -based SSA (n Figure 3b and 3c), if compared to UV-MFRSR and VIS-MFRSR (in Figure 3a). Could you please discuss the potential sources of absolute difference as well. For instance, your points 1 and 2 would both contribute so that MFRSR SSA is larger than AERONET SSA. Now, the scatter includes mainly points when MFRSR SSA is smaller than AERONET SSA. So a quantitative discussion about the possible sources of systematic biases, which differ

between these measurements, would be helpful for the reader to better understand not only the scatter, but also the mean overall differences.

Related to the above point and to your first point (fractional clouds). Would you see a reduced scatter between MFRSR and AERONET SSA, if you narrowed the 32 minutes averaging window? Given your arguments there, it should happen, so perhaps the role of this effect can be estimated?

SPECIFIC COMMENTS:

Figure 7, you list the wavelengths there in the caption, 673/675nm is missing.

---

## Referee Comment (RC2) · Anonymous Referee #3 · 6 Feb 2018

**GENERAL COMMENTS**

The paper by Mok et al. focuses on the comparison of aerosol single scattering albedo (SSA) retrieved by SKYNET (POM-02) and by a combination of instruments (AERONET, MFRSR and Pandora). The broad spectral range, including the ultraviolet band, covered by the comparison make this study original. Surface albedo is found to be one of the main sources of discrepancy (underestimation) in SKYNET compared to AMP.

The paper covers a very interesting research topic and is generally well written. I recommend the publication on AMT after addressing the following minor issues.

**SPECIFIC COMMENTS**

[Figure]

I have two remarks about the internal consistency of AMP retrievals.

1. Equation 1: in principle, to preserve consistency within the AMP triad, the gaseous optical depths used in MFRSR retrievals (tau_R, tau_NO2 and tau_O3), included in the right-hand side of Eq. 1, should be the same as the ones used by AERONET for the retrieval of the aerosol optical depth (tau_a) from the measurements of total optical atmospheric depth. Otherwise, slight differences in NO2 or O3 concentrations, pressure or used cross-sections could introduce some noise or fictitious biases (especially in the UV-VIS part of the spectrum). Can you discuss this point?

2. At page 7, the authors affirm that PDS retrievals from AERONET (which accounts for non-spherical aerosols) are used to calculate SSA from MFRSR assuming spherical particles. Isn't it an inconsistency? The authors should explain that most aerosol are spherical at the measuring site or that non-spherical aerosol were excluded from the analysis (e.g., based on some AERONET output parameters).

On a different note, do the authors have an explanation why they do not find the SSA overestimation as the previous studies at VIS and IR ranges? Since emphasis is laid on this contrast with the previous literature (e.g., page 10 l. 4-5 and l. 21-23), some explanations should be provided.

Finally, I would suggest to expand the conclusions, e.g. by including a special remark for terrains covered by snow and recommendations on how to determine the optimal surface albedo to be used in SKYNET inversions if no other co-located instrument is available at a specified measuring station.

TECHNICAL CORRECTIONS

page 1 title: the title refers to "spectral aerosol absorption" without mentioning explicitly the "single scattering albedo", which is the main topic of the paper and the only physical quantity provided as a result (apart from AOD and Angstrom exponent). I would suggest to change the title accordingly and not to mention in the abstract the quantities

that are not directly discussed in the paper (column effective imaginary refractive index (k) and aerosol absorption optical depth (AAOD));

page 2 l. 1-16: this first paragraph puts together too many topics that should be dealt with separately (radiative effects - consisting in scattering and absorption (not only absorption), health effects, photochemical smog, etc.). The result is a bit confusing for the reader and somehow disconnected from the main topic of the paper. I would suggest to rewrite this whole paragraph;

page 2 l. 15: "in the UV remain one of the most difficult tasks..." -> this is a key point. Explain why it is a difficult task;

page 3 l. 23: "equipped with" -> "mounted on" or "fitted to";

page 5 Eq. 1: the equation should be introduced by a sentence;

page 5 l. 29: "second order polynomial interpolation/extrapolation least-squares fit in logarithmic space..." -> replace this complex sentence with a formula;

page 5 l. 31: "a Pandora" -> "Pandora";

page 6 l. 3: "from the OMI" -> include a link to the data or explain which product was used;

page 6 l. 32: "either from MFRSR... or AERONET" -> explain how either one or the other quantity is chosen;

page 7 l. 23: "the static calibration" -> "the so-called static calibration";

page 7 l. 25: "use dynamic ... method" -> "use the dynamic ... method";

page 7 l. 27-28: "during very hot summer" -> does this mean that the agreement is better in the cold season because of lower temperature? Also, I do not understand how the daily temperature variations (line 27) can be taken into account using a two-month average period (page 8);

[Figure]

page 8 l. 1-2: "to minimize the temporal stability" -> "to account for the temporal variability" ? "consider the consistency with the above-mentioned static calibration constants" -> what do you mean? Could you rephrase?

page 8 l. 6: "the known field of view of the instrument" -> this seems to be a key point from previous literature. Could you explain what method you used to determine the FOV?

page 8 l. 21: "UV- and VIS-MFRSR retrieved SSA at 440 nm" -> "SSA retrieved at 440 nm by the UV- and VIS-MFRSR instruments";

page 9 l. 14: "Comparing" -> "Compared to the";

page 9 l. 20: "NO2 that is not completely accounted for in the AERONET retrievals" -> explain why;

page 10 l. 18-23: are these lines a typo? They are a repetition of the previous paragraph;

page 11 l. 12: "significantly increases the SSA (by 0.01)" -> how can a 0.01 increase be defined "significant"? Same at line 15: "significantly";

page 11 l. 19: "is a critical pre-condition" -> then, since this is a pre-condition, why not move this section before the SSA discussion?

page 11 l. 28-29: "gaseous absorption ... not taken into account in the sky radiances ... inverted in the AERONET Version 2 retrievals" -> could you add a bibliographic reference about this issue?

---

## Author Comment (AC1) · 6 Mar 2018

**amt-2017-380**

**Author response to reviews**

Mok et al., "Comparisons of spectral aerosol absorption in Seoul, South Korea"

[Reviewer comments are in black, responses in red]

Anonymous Referee #1

This manuscript by Mok et al., "Comparisons of spectral aerosol absorption in Seoul, South Korea", presents a comparison of SKYNET-retrieved SSA in the UV with the SSA derived from a combination of AERONET, MFRSR, and Pandora retrievals in Seoul, South Korea in spring and summer of 2016. There have been only a limited number of measurements / measurement campaigns focusing on absorption at UV wavelengths, therefore, the topic of this study is of great interest and relevance. The scope of the paper is both concise and specific, and my minor comments are mainly related to the need to clarify some of the issues.

We thank the reviewer for the positive assessment and summary

Before publication, the following points should be addressed:

GENERAL COMMENTS:

I was missing some more information and details about the measurements, for instance

regarding the following two points.

1. Figure 5 (and 7) shows a large variability for each SKYNET mean value, so apparently it is shown based on several measurements within 32 minutes, but what is the temporal resolution of SKYNET measurements? I think this was never mentioned.

Like the AMP retrievals, we used the same temporal resolution of the SKYNET measurements, which are averaged within ±16 minutes from the AERONET retrieval time for consistency.

For the clarification, we added the following statements at Page 3, Line 22:

"are retrieved every 10 minutes using standard processing software SKYRAD.pack"

The data shown in Fig. 5 and Fig. 7 are ±16 minute averages around AERONET inversion time for both MFRSR and SKYNET.

We clarified in the caption of Figure 5:
"Figure 5. Comparisons of AMP-retrieved with SKYNET-retrieved SSA (±16 minute average)"

2. Page 9, Line 25, here solar aureole corrections are mentioned. Please include few sentences to explain this correction in some detail. Also, this same issue applies also, to some extent, to Cimel measurements (diffuse light in FOV). Could you discuss the relative importance of this kind of uncertainty in both measurements?

We agree with suggestion. We added the following sentences on page 9 after line 26:

"The aureole correction is less important to the AERONET measurements because of the small FOV ~ 1.2$^{o}$ (Sinyuk et al., 2012) than to the shadowing measurements from MFRSR (Krotkov et al., 2005a). The empirical MFRSR aureole correction (Harrison et al., 1994) tends to underestimate aureole contribution to the diffuse irradiance for coarse aerosol particles and cirrus clouds (Min et al., 2004; Yin et al., 2015)."

Sinyuk, A., Holben, B. N., Smirnov, A., Eck, T. F., Slutsker, I., Schafer, J. S., Giles, D. M. and Sorokin, M.: Assessment of error in aerosol optical depth measured by AERONET due to aerosol forward scattering, Geophys. Res. Lett., 39, L23806, doi:10.1029/2012GL053894, 2012.

Min, Q. L., Joseph, E. and Duan, M.: Retrievals of thin cloud optical depth from a multifilter rotating shadowband radiometer, J. Geophys. Res., 109, D02201, doi:10.1023/2003JD003964, 2004.

Yin, B., Min, Q. and Joseph, E.: Retrievals and uncertainty analysis of aerosol single scattering albedo from MFRSR measurements, J. Quant. Spectrosc. Radiat. Transf., 150, 95-106, doi:10.1016/j.jqsrt.2014.08.012, 2015.

3. The focus of the paper is on UV wavelengths and thus one should not perhaps concentrate too much on the longer wavelengths, however I cannot help wondering about the comparison in the Figure 5 and at the wavelengths > 400nm. Now the explanation was given that the larger scatter is due to the lower AOD and related larger uncertainty. However, it is not only the larger scatter, but also the systematic behavior that stands out, e.g. at 500 nm for largest AOD (red points) SKYNET is showing very little SSA variability. SKYNET value is close to 0.93, when AERONET SSA varies from 0.88 to 0.98. AOD is not very low in these cases at 500nm, when it is larger than 0.4 at 440nm. Is there any idea why this happens?

As a reviewer mentioned, the $SSA_{500}$(SKYNET) values are ~0.909 to ~0.968, whereas the $SSA_{500}$(AMP) values are ~0.883 to ~0.975 in Figure 5(d), thus SKYNET SSA values show less variation compared to AMP SSA at 500 nm. However, the amount of data for comparison with $AOD_{440} \geq 0.4$ is only 19. Therefore, it is difficult to determine if this variation is significant with such a small number of data points. For example, if we change the AOD threshold from 0.4 to 0.3 and comparisons have more matchup samples, the SKYNET SSA shows similar variations as below.

[Figure]

(upper panel) original Figure 5(d). Red dots are filtered using $AOD_{440} \geq 0.4$.

(lower panel) same with Figure 5(d) except red dots are filtered using $AOD_{440} \geq 0.3$

Similar pattern and poor agreement seems to be true also at 675nm. Also, the vertical error bars are sometimes strikingly large. In your Figure 8 you show that AODs match well, so what could be the main reason to cause a variability this large SSA variability within a short period of measurements?

The horizontal bars in Fig. 5 and 7 show estimated uncertainties of the AMP SSA mean values (i.e. excluding natural variability) within ±16 minute time window. Because SKYNET retrievals do not provide SSA uncertainties for the individual retrievals, the vertical bars show one standard deviation of the SKYNET retrieved individual SSA values within ±16 minute time window (i.e. including natural variability). Natural SSA short-term variability makes vertical bars typically larger.

We added clarification in the captions of Fig. 5 and Fig. 7.

"The horizontal bars show estimated uncertainties of the AMP SSA mean values (i.e. excluding natural variability) within ±16 minute time window. The vertical bars show one standard deviation of the SKYNET retrieved individual SSA values within ±16 minute time window (i.e. including natural variability)."

4. Page 9, Lines 14-26. Here are several possible explanations given for a larger scatter between AERONET and MFRSR-based SSA (Figure 3b and 3c), if compared to UV-MFRSR and VIS-MFRSR (in Figure 3a). Could you please discuss the potential sources of absolute difference as well. For instance, your points 1 and 2 would both contribute so that MFRSR SSA is larger than AERONET SSA. Now, the scatter includes mainly points when MFRSR SSA is smaller than AERONET SSA. So a quantitative discussion about the possible sources of systematic biases, which differ between these measurements, would be helpful for the reader to better understand not only the scatter, but also the mean overall differences.

We agree with suggestion. We added the following quantitative discussion at the end of page 9:

"The aureole correction is less important to the AERONET measurements because of the small FOV ~ 1.2° (Sinyuk et al., 2012) than to the shadowing measurements from MFRSR (Krotkov et al., 2005a). The empirical MFRSR aureole correction (Harrison et al., 1994) tends to underestimate the aureole contribution to the diffuse irradiance for coarse aerosol particles and cirrus clouds (Min et al., 2004; Yin et al., 2015). The aureole undercorrection causes systematic underestimation of the diffuse irradiance and retrieved SSA by the MFRSR. Quantitatively, the bias varies for different locations: e.g., from +0.004 at the Santa Cruz, Bolivia (Mok et al., 2016) to -0.005 in Greenbelt, Maryland with fine mode dominated aerosols (Krotkov et al., 2009). We estimate that aureole SSA bias should be less than ~0.01 at Seoul."

5. Related to the above point and to your first point (fractional clouds). Would you see a reduced scatter between MFRSR and AERONET SSA, if you narrowed the 32 minutes averaging window? Given your arguments there, it should happen, so perhaps the role of this effect can be estimated?

Although the MFRSR can provide 1-miniute retrievals of SSA, the AERONET standard (Dubovik) algorithm requires 32 minutes of the almucantar scan time to retrieve SSA. So, it is not possible to narrow the 32 minutes averaging window for comparison between MFRSR and AERONET SSA.

SPECIFIC COMMENTS:

6. Figure 7, you list the wavelengths there in the caption, 673/675nm is missing.

We do not include 675 nm in the caption on purpose. As L4 in Page 11, the spectrally invariant SKYNET-assumed surface albedo~0.1 is close to the AERONET surface albedo at 675 nm (Figure 6). Thus, we do not re-process the SKYNET inversion at 675 nm.

For the clarification, we added the following sentence in the caption of Figure 7:
SKYNET SSA at 675 nm is the same with Figure 5(e).

---

## Author Comment (AC2) · 6 Mar 2018

**amt-2017-380**

**Author response to reviews**

Mok et al., "Comparisons of spectral aerosol absorption in Seoul, South Korea"

[Reviewer comments are in black, responses in red]

Anonymous Referee #3

GENERAL COMMENTS

The paper by Mok et al. focuses on the comparison of aerosol single scattering albedo (SSA) retrieved by SKYNET (POM-02) and by a combination of instruments (AERONET, MFRSR and Pandora). The broad spectral range, including the ultraviolet band, covered by the comparison make this study original. Surface albedo is found to be one of the main sources of discrepancy (underestimation) in SKYNET compared to AMP.

The paper covers a very interesting research topic and is generally well written. I recommend the publication on AMT after addressing the following minor issues.

We thank the reviewer for the positive assessment and summary

SPECIFIC COMMENTS

I have two remarks about the internal consistency of AMP retrievals.

1. Equation 1: in principle, to preserve consistency within the AMP triad, the gaseous optical depths used in MFRSR retrievals (tau_R, tau_NO2 and tau_O3), included in the right-hand side of Eq. 1, should be the same as the ones used by AERONET for the retrieval of the aerosol optical depth (tau_a) from the measurements of total optical atmospheric depth. Otherwise, slight differences in NO2 or O3 concentrations, pressure or used cross-sections could introduce some noise or fictitious biases (especially in the UV-VIS part of the spectrum). Can you discuss this point?

During MFRSR calibration, we correct AERONET AOD to account for differences between measured and climatological $NO_2$, ozone, and surface pressure values, making AMP retrievals internally consistent (Krotkov et al., 2009). We compare AERONET climatology with actual Pandora measurement in Seoul and see large underestimation (up to a factor of ~2) for high polluted episodes. Combining AERONET, MFRSR, and Pandora (AMP) retrievals ensures most accurate partitioning between aerosol and gaseous absorption, although this is not yet possible for all of ~400 AERONET sites.

2. At page 7, the authors affirm that PSD retrievals from AERONET (which accounts for non-spherical aerosols) are used to calculate SSA from MFRSR assuming spherical particles. Isn't it an inconsistency? The authors should explain that most aerosol are spherical at the measuring site or that non-spherical aerosol were excluded from the analysis (e.g., based on some AERONET output parameters).

We acknowledge the inconsistency of assuming spherical particles in MFRSR retrievals. Below Figure shows similar comparison (Figure 3b) for cases with AERONET sphericity exceeding 95%. We see similar results but this leads to much smaller statistical sample size, not allowing us to compare with SKYNET SSA retrievals.

[Figure]

On a different note, do the authors have an explanation why they do not find the SSA overestimation as the previous studies at VIS and IR ranges? Since emphasis is laid on this contrast with the previous literature (e.g., page 10. 4-5 and 21-23), some explanations should be provided.

The lack of overestimation is due, at least partly, to improved quality checks for the solar disk scan data used to determine FOV (referred to as SVA in previous literatures). In addition, the present study uses a slightly different approach for the determination of the calibration constant $<F_0>$. As mentioned in Section 3.3, while daily $<F_0>$ values for entire UV-VIS-NIR channels have not been given in previous studies, we think that reanalysis of their observation data by this approach is preferable to confirm the consistency.

We added the following statements at L5 in P10:

"Differently from previous studies, we found that average SKYNET SSA is in good agreement with average AMP SSA at VIS and NIR ranges (Figure 5 and Table 3). This is at least partly because we used the improved quality checks for the solar disk scan data used to determine the FOV. In addition, we used daily $<F_0>$ values for entire UV-VIS-NIR channels have not been given in previous studies (See details in Section 3.3)."

3. Finally, I would suggest to expand the conclusions, e.g. by including a special remark for terrains covered by snow and recommendations on how to determine the optimal surface albedo to be used in SKYNET inversions if no other co-located instrument is available at a specified measuring station.

We agree with reviewer's suggestions.

Since the surface albedo has a significant impact on SSA retrievals, future studies relevant to SKYNET inversions might determine the optimal surface albedo from the MODIS climatology (Moody et al., 2008) combined with bidirectional reflectance distribution function (BRDF) models to account for change as a function of solar zenith angle, like AERONET inversions.

In the Version 3 database the AERONET input for surface reflectance is based on the BRDF determined from MODIS data (V005 product) for all locations as described in:

Wang, Z., Schaaf, C. B., Sun, Q., Shuai, Y. and Román, M. O.: Capturing rapid land surface dynamics with Collection V006 MODIS BRDF/NBAR/Albedo (MCD43) Products, Remote Sens. Environ., 207, 50–64, doi: 10.1016/j.rse.2018.02.001, 2018.

In presence of snow and ice, the global daily surface albedo from the National Snow and Ice Data Center can be used. However, the snow/ice albedos have very high uncertainty due to very dynamic nature of snow and ice reflectance. This will be addressed in our future paper.

We added the following sentence in conclusion (L8 in P13) as reviewer suggested.

"Future studies relevant to SKYNET SSA inversions might determine the optimal surface albedo from the MODIS climatology (Moody et al., 2008) and/or combined with BRDF models (Wang et al., 2018) if no other co-located instrument is available."

TECHNICAL CORRECTIONS

4. page 1 title: the title refers to "spectral aerosol absorption" without mentioning explicitly the "single scattering albedo", which is the main topic of the paper and the only physical quantity provided as a result (apart from AOD and Angstrom exponent). I would suggest to change the title accordingly and not to mention in the abstract the quantities that are not directly discussed in the paper (column effective imaginary refractive index (k) and aerosol absorption optical depth (AAOD));

We agree with suggestion. We changed the title as

"Comparisons of spectral aerosol single scattering albedo in Seoul, South Korea"

Also, we removed column effective imaginary refractive index (k) and AAOD in the abstract as:

"Measurements of column average atmospheric aerosol single scattering albedo (SSA) are performed on the ground by the NASA AERONET in the visible (VIS) and near-infrared (NIR) wavelengths and in the UV-VIS-NIR by the SKYNET networks."

5. page 2. 1-16: this first paragraph puts together too many topics that should be dealt with separately (radiative effects - consisting in scattering and absorption (not only absorption), health effects, photochemical smog, etc.). The result is a bit confusing for the reader and somehow disconnected from the main topic of the paper. I would suggest to rewrite this whole paragraph;

We agree with suggestion. We rewrite this paragraph to show clear message of the main topic of this paper to the reader as below.

"Aerosols affect both the surface and outgoing radiation affecting Earth's radiative balance. To quantify the radiative effects of aerosols, the aerosol optical depth (AOD) and single scattering albedo (SSA) are monitored using ground-based, orbital and sub-orbital platforms. The potential climate effects of absorbing aerosols have received considerable attention lately (Myhre et al., 2013). In addition to climatic effects, aerosol absorption effects on surface UV irradiance and photolysis rates have important implications for tropospheric photochemistry, human health, and agricultural productivity (Dickerson et al., 1997; Krotkov et al., 1998; He and Carmichael, 1999; Castro et al., 2001; Mok et al., 2016). Measurements of column atmospheric aerosol absorption and its spectral dependence in the UV remain one of the most difficult tasks in atmospheric radiation measurements due to the lack of co-incident measurements of aerosol and gaseous absorption properties in the UV."

6. page 2. 15: "in the UV remain one of the most difficult tasks..." -> this is a key point.

Explain why it is a difficult task;

Compared to longer visible and NIR wavelengths, the gaseous absorption of ozone and $NO_2$ becomes important when trying to retrieve the column aerosol absorption in the UV. This problem occurs because there are lack of co-incident measurements of aerosol and gaseous absorption properties in the UV.

We changed the statements:

"Measurements of column atmospheric aerosol absorption and its spectral dependence in the UV remain one of the most difficult tasks in atmospheric radiation measurements due to the lack of co-incident measurements of aerosol and gaseous absorption properties in the UV."

7. page 3. 23: "equipped with" -> "mounted on" or "fitted to";

We agree with suggestion:

"The ability for UV (340 and 380 nm) channels mounted on the PREDE POM-02 sky radiometer used by SKYNET is investigated in this study."

8. page 5. Eq. 1: the equation should be introduced by a sentence;

We agree with suggestion. We changed the location of this sentence to L21 in P5 to show the equation is introduced by a sentence as below.

We use an estimate of the calibration constant for each individual 1-minute MFRSR measurement at each wavelength (i.e., extraterrestrial voltage, $V_0(\lambda,t)$) calculated using equation (1) to normalize measured direct and diffuse voltages (same calibration in shadowing technique) and as a quality assurance tool to retain only the best quality measurements consistent with the AERONET AOD measurements.

$$\ln V_0(\lambda,t) = \ln(V_{dim}(\lambda,t)) + \sec(SZA(t))\,[\tau_a(\lambda,t) + \tau_R(\lambda,t) + \tau_{NO_2}(\lambda,t) + \tau_{O_3}(\lambda,t)\,]\,, \tag{1}$$

9. page 5. 29: "second order polynomial interpolation/extrapolation least-squares fit in logarithmic space..." -> replace this complex sentence with a formula;

We agree with suggestion. We changed the statement by adding a formula as below.

"$\tau_a(\lambda,t)$ is gaseous corrected and spectrally interpolated/extrapolated AOD to the MFRSR wavelengths applying a least-squares fit of the equation ($\ln \tau_a = a_0 + a_1 \ln \lambda + a_2 (\ln \lambda)^2$) (Eck et al., 1999) using AERONET spectral level 2 AOD"

10. page 5. 31: "a Pandora" -> "Pandora";

We agree with suggestion:

"For cases when $NO_2$ and $O_3$ values are not available from Pandora spectrometer,"

11. page 6. 3: "from the OMI" -> include a link to the data or explain which product was used;

We agree with suggestion:

"For cases when $NO_2$ and $O_3$ values are not available from Pandora spectrometer, satellite $NO_2$ (OMNO2 L2 v3.0) and ozone (OMTO3 L2 v8.5) measurements from the OMI are used (data are available at http://avdc.gsfc.nasa.gov under the Aura sub-menu)."

12. page 6. 32: "either from MFRSR... or AERONET" -> explain how either one or the other quantity is chosen;

We can manually choose which AOD retrievals (MFRSR or AERONET) are used for the AMP SSA inversion. In this study, we only used gaseous corrected AERONET AOD for consistency.

We added the statement to L1 in P7:

"In this study, we only used gaseous corrected AERONET AOD for consistency."

13. page 7. 23: "the static calibration" -> "the so-called static calibration";

We agree with suggestion:

"The first approach is to use the so-called static calibration constants."

14. page 7. 25: "use dynamic ... method" -> "use the dynamic ... method";

We agree with suggestion:

"The second approach is to use the dynamic on-site calibration method, based on the Improved Langley method (Campanelli et al., 2007; Khatri et al., 2016)."

15. page 7. 27-28: "during very hot summer" -> does this mean that the agreement is better in the cold season because of lower temperature? Also, I do not understand how the daily temperature variations (line 27) can be taken into account using a two-month average period (page 8);

Yes, it means that the agreement is better in the cold season because of lower temperature. However, this would not be the case, when the temperature is too low. Since the present study focuses on the season from spring to summer, we state "during very hot summer for instance".

We also agree with the reviewer that the daily temperature variations cannot be taken into account.

Accordingly, the sentence has been rewritten as "... on a monthly time scale ...".

16. page 8. 1-2: "to minimize the temporal stability" -> "to account for the temporal variability" ? "consider the consistency with the above-mentioned static calibration constants" -> what do you mean? Could you rephrase?

Considering the reviewer's comments, the sentence has been rephrased to

"To account for the temporal variability of $<F_0>$ by ±1–3% caused by temperature variation, the following method was used in this study."

17. page 8. 6: "the known field of view of the instrument" -> this seems to be a key point from previous literature. Could you explain what method you used to determine the FOV?

The FOV was determined by the solar disk scan method (Nakajima et al., 1996; Uchiyama et al., 2018). For the present study, careful quality check for the solar disk scan data was made by identifying and excluding apparent low-quality data, in which the measured normalized intensity showed an unexpected increase as the scattering angle increases.

Nakajima, T., Tonna, G., Rao, R., Boi, P., Kaufman, Y. and Holben, B.: Use of sky brightness measurements from ground for remote sensing of particulate polydispersions, Appl. Opt., 35, 2672–2686, doi:10.1364/AO.35.002672, 1996.

Uchiyama A., Matsunaga, T. and Yamazki, A.: The instrument constant of sky radiometers (POM-02), Part II: Solid view angle, Atmos. Meas. Tech. Discuss., doi:10.5194/amt-2017-432, 2018.

We added the following statement at L4 in P8:

"Assuming the field of view (FOV) of the SKYNET instrument is known by the solar disk scan method (Nakajima et al., 1996; Uchiyama et al., 2018),"

18. page 8. 21: "UV- and VIS-MFRSR retrieved SSA at 440 nm" -> "SSA retrieved at 440 nm by the UV- and VIS-MFRSR instruments";

We agree with suggestion:

"First, the individual 1-minute SSA retrieved at 440 nm ($SSA_{440}$) by the UV- and VIS-MFRSR instruments are compared to demonstrate the high degree of consistency for a combined set of modified UV- and VIS-MFRSR instruments (Figure 3a)."

19. page 9. 14: "Comparing" -> "Compared to the";

We agree with suggestion:

"Compared to the low scatter in $SSA_{440}$ differences between UV-MFRSR and VIS-MFRSR (Figure 3a), Figures 3b and 3c show larger scatter between either UV-MFRSR (Figure 3b) or VIS-MFRSR (Figure 3c) and AERONET $SSA_{440}$."

20. page 9. 20: "NO2 that is not completely accounted for in the AERONET retrievals" -> explain why;

AERONET Version 2 AOD measurements are corrected for $NO_2$ absorption using monthly average satellite climatologies from SCIAMACHY satellite retrievals (https://aeronet.gsfc.nasa.gov/new_web/Documents/version2_table.pdf). However, $NO_2$ absorption is not taken into account in the sky radiances that are inverted in the AERONET SSA inversion (Dubovik) algorithm in Version 2.

21. page 10. 18-23: are these lines a typo? They are a repetition of the previous paragraph;

Considering the reviewer's comments, we removed this paragraph.

22. page 11. 12: "significantly increases the SSA (by 0.01)" -> how can a 0.01 increase be defined "significant"? Same at line 15: "significantly";

Considering the reviewer's comments, we remove "significantly".

23. page 11. 19: "is a critical pre-condition" -> then, since this is a pre-condition, why not move this section before the SSA discussion?

We think that to discuss possible factors for discrepancy between the AMP and SKYNET SSA in one section (Section 4.3) is better way for readers to understand like Khatri et al. (2016) did.

Khatri, P., Takamura, T., Nakajima, T., Estellés, V., Irie, H., Kuze, H., Campanelli, M., Sinyuk, A., Lee, S.-M., Sohn, B. J., Pandithurai, G., Kim, S.-W., Yoon, S. C., Martinez-Lozano, J. A., Hashimoto, M., Devara, P. C. S. and Manago, N.: Factors for inconsistent aerosol single scattering albedo between SKYNET and AERONET, J. Geophys. Res.-Atmos., 121, 1859-1877, doi:10.1002/2015JD023976, 2016.

24. page 11. 28-29: "gaseous absorption ...  not taken into account in the sky radiances... inverted in the AERONET Version 2 retrievals" -> could you add a bibliographic reference about this issue?

There is no bibliographic reference that discusses this issue. When papers in the early stages of preparation for the new AERONET Version 3 database are available in the future, this issue will be discussed.

---

## Editor Comment (EC1) · S. Kazadzis (Editor) · 7 Mar 2018

Dear authors,

The paper is very interesting and present unique results and methodology for a scientific area (columnar UV absorption) that currently only limited information can be found in the literature.

One aspect that the first reviewer is bringing up is the observed differences for SSA comparisons for higher wavelengths. As stated the paper is showing a method for the UV region. However, it is a bit puzzling that somehow this comparison provides an evaluation of the method using skynet SSA@UV retrievals. However the more "standard" SSA retrievals at VIS and NIR wavelengths show larger descrepencies. Even

compared with the SSA@340nm that AOD differences should lead to SSA deviations (as already stated by the authors).

So to my point of view this issue needs further clarification in order for the reader to understand if the differences comparing Skynet SSA retrievals and SSA from the two methods (the standard cimel and the AMP synergistic one) for the VIS-NIR and the UV respectively, are due to instrument or method related differences.

regards SK

---

## Author Response (AR2)

**amt-2017-380**

**Authors' combined response file**

Dear Dr. Stelios Kazadzis,

Following our interactive discussion, our paper is now revised for your consideration for publication in Atmospheric Measurement Techniques.

We have included below response to the editor and a mark-up manuscript showing the changes made in response to editor's suggestions.

All co-authors agree with the changes and this final version.

I look forward to your decision.

Yours sincerely,

Jungbin Mok

**amt-2017-380**

**Response to the editor Dr. Stelios Kazadzis**

Mok et al., "Comparisons of spectral aerosol absorption in Seoul, South Korea"

[responses in red]

5  Associate Editor Decision: Publish subject to minor revisions (review by editor) (19 Mar 2018) by Stelios Kazadzis

This is a very interesting paper with a lot of new findings, including methods, results, comparisons related with SSA retrieval.

We thank the associate editor for the positive assessment

10  I only have two comments:

1. Since you are (partly with one figure) presenting the spectral dependence of SSA from VIS-NIR to the UV it would be good to discuss in the introduction other studies with similar findings at urban areas, for example:

Ialongo, I., Buchard, V., Brogniez, C., Casale, G. R., and Siani, A. M.: Aerosol Single Scattering

15  Albedo retrieval in the UV range: an application to OMI satellite validation, Atmos. Chem. Phys.,10, 331–340, doi:10.5194/acp-10-331-2010, 2010.

Kazadzis, S., Raptis, P., Kouremeti, N., Amiridis, V., Arola, A., Gerasopoulos, E., and Schuster, G. L.: Aerosol absorption retrieval at ultraviolet wavelengths in a complex environment, Atmos. Meas. Tech., 9, 5997-6011, doi:10.5194/amt-9-5997-2016, 2016.

20  We agree with suggestion. We added the following sentence at L16, P2 in the introduction and added references.

"Similarly, the enhancement of aerosol absorption at UV wavelengths was observed at urban cities such as Rome, Italy (Ialongo et al., 2010) and Athens, Greece (Kazadzis et al., 2016), especially in winter."

25  2. Secondly: Following the comment of one of the reviewers on the discrepancies of the two retrievals (AMP, SKYNET) at higher wavelengths:

I think that the discrepancies (fig. 5 and 7) for 675nm and 870nm are relatively quite high with cc 0.55 and 0.41, at least compared the VIS range. But also compared to the UV range where someone should

expect larger discrepancies mainly due to the AOD differences at 340nm. I think it is important to discuss this issue in the conclusion section so future readers or users of such data can use your assessment in order to describe their results. Especially because of the fact that SSA retrieval with those instruments at those wavelengths are methodologically more standard in the literature.

5 We discussed this issue with SKYNET co-authors and added the following statement in the conclusions:

"The relatively poor correlations between AMP and SKYNET SSA at 675 and 870 nm compared to shorter wavelengths should reflect, at least partly, the fact that AODs at 675 and 870 nm were much lower than AODs at other shorter wavelengths. The second issue is smaller Rayleigh scattering, which

10 greatly reduces diffuse sky irradiance and causes larger noise in diffuse to direct ratio. Future studies using more observations with higher AODs are needed to better quantify SSA at 675 and 870 nm."

S. Kazadzis (Editor)

stelios.kazadzis@pmodwrc.ch

Dear authors,

The paper is very interesting and present unique results and methodology for a scientific area (columnar UV absorption) that currently only limited information can be found in the literature.

20 One aspect that the first reviewer is bringing up is the observed differences for SSA comparisons for higher wavelengths. As stated the paper is showing a method for the UV region. However, it is a bit puzzling that somehow this comparison provides an evaluation of the method using skynet SSA@UV retrievals. However the more "standard" SSA retrievals at VIS and NIR wavelengths show larger discrepancies. Even compared with the SSA@340nm that AOD differences should lead to SSA

25 deviations (as already stated by the authors).

So to my point of view this issue needs further clarification in order for the reader to understand if the differences comparing Skynet SSA retrievals and SSA from the two methods (the standard cimel and

the AMP synergistic one) for the VIS-NIR and the UV respectively, are due to instrument or method related differences.

We agree with editor's comments. Please see the response to comment #2 above.

[revised manuscript text omitted]